# Participation in the nationwide cervical cancer screening programme in Denmark during the COVID-19 pandemic: An observational study

**Tina Bech Olesen[1]\*, Henry Jensen[1], Henrik Møller[1], Jens Winther Jensen[1], Marianne Waldstrøm[2,3], Berit Andersen[4,5]**

[1]The Danish Clinical Quality Program - National Clinical Registries, Aarhus, Denmark; [2]Department of Pathology, Lillebaelt Hospital, Vejle, Denmark; [3]Department of Regional Health Research, University of Southern Denmark, Odense, Denmark; [4]University Research Clinic for Cancer Screening, Department of Public Health Programmes,Randers Regional Hospital, Aarhus, Denmark; [5]Department of Clinical Medicine, Aarhus University, Aarhus, Denmark

**\*For correspondence:** forthemanuscripts@gmail.com

**Competing interest:** The authors declare that no competing interests exist.

## Abstract

**Background:** In contrast to most of the world, the cervical cancer screening programme continued in Denmark throughout the COVID-19 pandemic. We examined the cervical cancer screening participation during the pandemic in Denmark.

**Methods:** We included all women aged 23–64 y old invited to participate in cervical cancer screening from 2015 to 2021 as registered in the Cervical Cancer Screening Database combined with population-wide registries. Using a generalised linear model, we estimated prevalence ratios (PRs) and 95% CIs of cervical cancer screening participation within 90, 180, and 365 d since invitation during the pandemic in comparison with the previous years adjusting for age, year, and month of invitation.

**Results:** Altogether, 2,220,000 invited women (in 1,466,353 individuals) were included in the study. Before the pandemic, 36% of invited women participated in screening within 90 d, 54% participated within 180 d, and 65% participated within 365 d. At the start of the pandemic, participation in cervical cancer screening within 90 d was lower (pre-lockdown PR = 0.58; 95% CI: 0.56–0.59 and first lockdown PR = 0.76; 95% CI: 0.75–0.77) compared with the previous years. A reduction in participation within 180 d was also seen during pre-lockdown (PR = 0.89; 95% CI: 0.88–0.90) and first lockdown (PR = 0.92; 95% CI: 0.91–0.93). Allowing for 365 d to participation, only a slight reduction (3%) in participation was seen with slightly lower participation in some groups (immigrants, low education, and low income).

**Conclusions:** The overall participation in cervical cancer screening was reduced during the early phase of the pandemic. However, the decline almost diminished with longer follow-up time.

**Funding:** The study was funded by the Danish Cancer Society Scientific Committee (grant number R321-A17417) and the Danish regions.

## Editor's evaluation

This article shows how the COVID-19 pandemic affected cervical cancer screening participation in the organized screening program of Denmark. Through registry data covering the entire population, the study shows that while short-term (90 days) participation after invitation dropped, long-term

(365 days) participation remained stable. These results will be of interest to public health specialists and researchers working on pandemic recovery efforts related to cancer screening worldwide.

## Introduction

The COVID-19 pandemic is a global health crisis, which has caused extensive disruptions to the society and to the healthcare systems across the world. Population-wide restrictions ('lockdowns') were imposed in most countries throughout the pandemic closing down schools and workplaces and restricting travel to reduce the transmission of COVID-19 and to limit the potential burden on the healthcare systems. Within the healthcare system, prioritisations and re-organisations were done to ensure sufficient capacity to take care of patients in need of hospitalisation due to COVID-19. The prioritisations within the healthcare system resulted in a temporary halting of the cervical cancer screening programme in most of the world. On the contrary, in Denmark, the cervical cancer screening programmes remained open throughout the pandemic. At the same time, however, at the national televised press conferences, the health authorities asked the Danish population to stay at home if possible, and concurrently, the Danish College of General Practitioners recommended general practitioners to postpone routine cervical smears during a 4 wk period in March/April 2020 (*Dansk Selskab for Almen Medicin, 2020*). Nevertheless, the cervical cancer screening programme continued – and invitations and reminders were sent out – throughout the pandemic in Denmark.

It is estimated that the disruptions to the cervical cancer screening programmes in high-income countries because of the pandemic could potentially increase cervical cancer cases by up to 5–6% and increase the number of cervical cancers detected at a higher stage (*Smith et al., 2021*). Disruptions to the cervical cancer screening programme may therefore be worrisome. Marked reductions in the number of women screened for cervical cancer during the early phase of the pandemic have been reported in many other countries (*Castanon et al., 2022*; *Cancer Registry of Norway, 2020*; *Meggetto et al., 2021*; *Ivanuš et al., 2021*), whilst the participation in cervical cancer screening during the pandemic in Denmark has not yet been described.

It is well known that participation in cervical cancer screening is generally reduced among women of lower socio-economic status (*Harder et al., 2018*) and among immigrant women (*Hertzum-Larsen et al., 2019*; *Badre-Esfahani et al., 2020*). This divergence in participation may have been exacerbated during the COVID-19 pandemic. However, so far no studies have put spotlight on this.

In this large, population-based nationwide study, we examined the participation in cervical cancer screening during the COVID-19 pandemic in Denmark in comparison with the previous years. Moreover, we examined whether the participation in cervical cancer screening during the pandemic differed across population groups with different socio-economic status.

## Methods
### Setting

The study was set in Denmark, which has a population of approximately 5.8 million inhabitants (*Statistics Denmark, 2021*). Denmark has a tax-funded healthcare system, with universal access to healthcare for all residents including national screening programmes for breast, cervical, and colorectal cancer. The population-based administrative and health registries in Denmark can be linked through the unique personal identifier assigned to all residents at birth or immigration (*Schmidt et al., 2014*, *Schmidt et al., 2019*).

### The cervical cancer screening programme

In Denmark, all women aged 23–64 y old are invited to participate in cervical cancer screening every 3 y (women aged 23–49 y old) or every 5 y (women aged 50–64 y old; *Bonde et al., 2022*). The women receive an invitation letter (electronic letters via secure digital e-mail since 2018; however, women exempted from digital mail still receive ordinary mail) with an invitation to book an appointment with their general practitioner for a cervical screening test. Reminders to participate in cervical cancer screening are sent out to non-participants after 3 mo and again after 6 mo. The obtained samples are analysed for cytology and/or HPV at a pathology department. The outcome of the test is sent to the woman and her general practitioner.

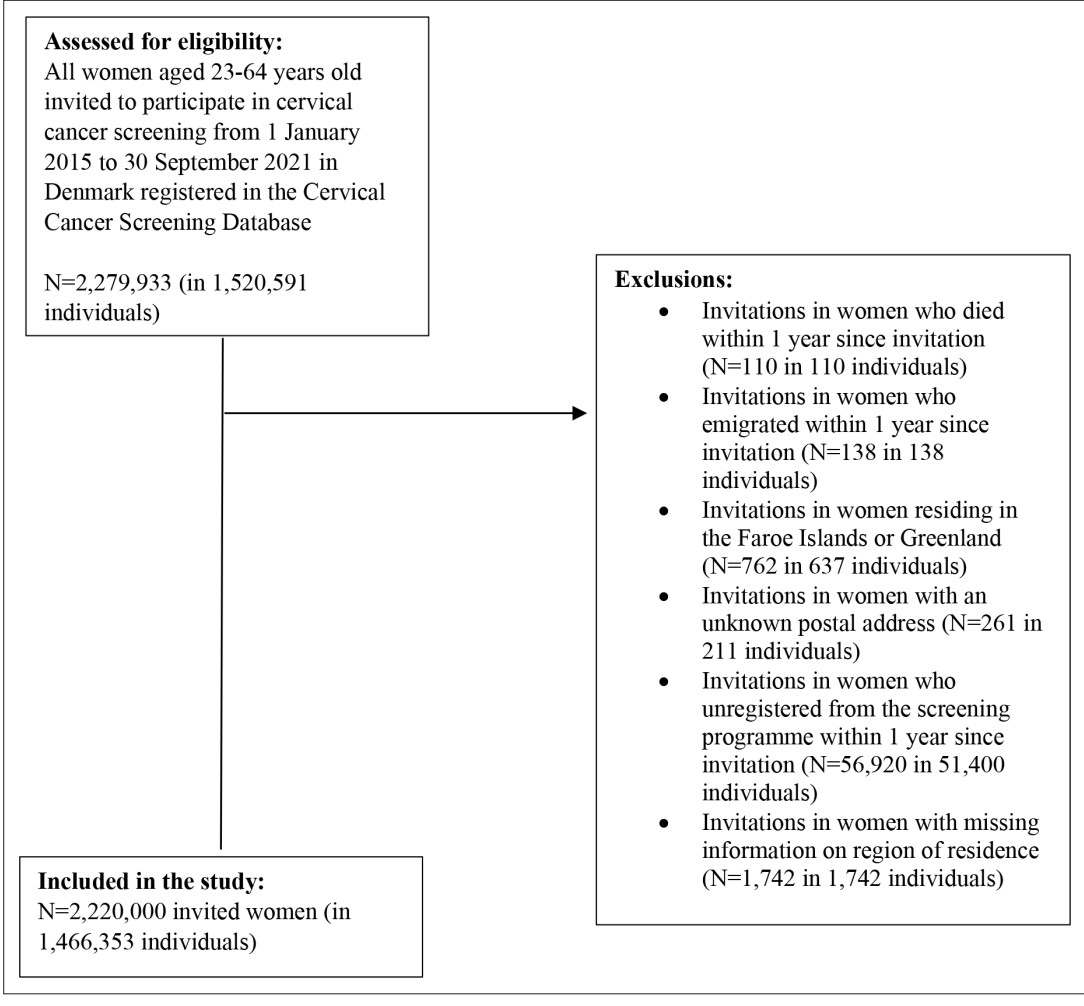

**Figure 1.** Flow-chart of the study population.

## The COVID-19 pandemic in Denmark

In Denmark, three main waves of the COVID-19 pandemic have occurred that is, in the spring of 2020, in the winter of 2020/2021, and again in the winter of 2021/2022 (*Statens Serum Institut, 2021a*).

In efforts to minimise the spread of the infection, population-wide restrictions ('lockdown') were imposed in Denmark 11 March 2020, and subsequently, large parts of the society were closed down. Within the healthcare system, elective procedures were cancelled or postponed, and resources were reallocated to take care of patients in need of hospitalisation because of COVID-19.

Extensive testing facilities were set up in Denmark from May 2020, providing COVID-19 tests free-of-charge to the whole population (*Pottegård et al., 2020*). In Denmark, laymen were trained to perform COVID-19 tests, which is in contrast to many other countries where healthcare personnel were allocated to perform COVID-19 tests. Vaccination against COVID-19 began in December 2020 in Denmark, and a high vaccination coverage has been achieved, and by March 2022, approximately 81% of the population had received two doses, and more than 61% had received three doses of the vaccine (*Statens Serum Institut, 2021b*).

## Study population

The study population comprised all women aged 23–64 y old invited to participate in cervical cancer screening from 1 January 2015 to 30 September 2021, as registered in the Cervical Cancer Screening Database (*Rygaard, 2016*), which contain information on all women invited to participate in cervical cancer screening in Denmark since 2009. The Cervical Cancer Screening Database comprises population data from the Civil Registration System (*Schmidt et al., 2014*), including all persons with a

permanent address in Denmark; cervical cancer cases are obtained from the Danish Cancer Register (*Gjerstorff, 2011*), cervical cytology samples are obtained from the Danish Pathology Register (*Bjerregaard and Larsen, 2011*), and information on invitations and reminders is obtained from the invitation registration system.

We excluded invitations in women who died within 1 y since invitation (N=110), women who emigrated within 1 y since invitation (N=138), women residing in the Faroe Islands or Greenland (N=762), women with an unknown postal address (N=261), women who unregistered from the screening programme within 1 y since invitation (N=56,920), and invitations in women with missing information on region of residence (N=1742; *Figure 1*).

### Exposure of interest

The exposure of interest was the COVID-19 pandemic in Denmark. The different phases of the pandemic were defined, in accordance with the governmental responses to the COVID-19 pandemic in Denmark, as follows:

- Pre-pandemic period: 1 January 2015 to 31 January 2020.
- Pre-lockdown period: 1 February to 10 March 2020.
- First lockdown: 11 March to 15 April 2020.
- First re-opening: 16 April to 15 December 2020.
- Second lockdown: 16 December 2020 to 27 February 2021.
- Second re-opening: 28 February 2021 to 30 September 2021 (end of inclusion period).

When examining each outcome of interest as stated below (participation within 90, 180, and 365 d since invitation), the end of inclusion period varied, that is, the end of inclusion was 31 December 2020 when examining participation in screening within 365 d since invitation meaning that participation within second lockdown could only be observed among women invited until 31 December 2020. To ensure at least 90 d complete follow-up on all tests, the data on cervical cytology samples covered up until 31 December 2021.

### Outcome of interest

The main outcome of interest was participation in cervical cancer screening defined as having a cervical cancer screening test performed within 90, 180, and 365 d since invitation, respectively, among women invited to participate in the cervical screening programme. We thus calculated the proportion of women participating in cervical cancer screening within 90, 180, and 365 d since invitation, respectively, among invited women.

### Explanatory variables

The following variables were examined independently: age, ethnicity, cohabitation status, educational level, disposable income, and healthcare usage. Age was defined at the date of invitation, as registered in the Cervical Cancer Screening Database (*Rygaard, 2016*). From *Statistics Denmark, 2021*, we obtained information on ethnicity, marital status, educational level, and level of income. Ethnicity was categorised as Danish descent, western immigrant, non-western immigrant, and descendants of immigrants. Cohabitation status was categorised as living alone, co-habiting/co-living, and married (i.e. married or registered partnership) in accordance with *Statistics Denmark, 2021*. Education level was defined in accordance with the International Standard Classification of Education (ISCED) of the United Nations Education, Scientific, and Cultural Organisation (UNESCO) into short (ISCED level 1–2), medium (ISCED level 3–5), and long (ISCED level 6–8; *Statistics Denmark, 2021*). Income was defined as official disposable income depreciated to 2015 level and categorised into five quintiles. To indicate the level of healthcare use by each patient, we counted the total number of contacts (comprising face-to-face, telephone, and e-mail consultations) to general practitioners, private practising medical specialists, physiotherapists, and chiropractors in the year for invitation as registered in the Danish National Health Service Register (*Andersen et al., 2011*), which contain information on visits to primary healthcare (e.g., general practitioners and medical specialists) in Denmark since 1990. We categorised healthcare usage into five quintiles of the data as rare (0–3 visits per year), low (4–6 visits per year), average (7–11 visits per year), high (12–18 visits per year), and frequent (≥19 visits per years).

Information on cohabitation status was only available from Statistics Denmark until the end of February 2021, whereas all other socio-economic variables were available until end of the study period.

## Statistical analyses

We examined characteristics of women invited to participate in cervical cancer screening during the study period. Thereafter, we examined the participation in cervical cancer screening within 90 d, 180 d, and 365 d since invitation among women invited to participate in screening per month and during the different phases of the pandemic overall and stratifying by the explanatory variables. Additionally, we examined time from invitation to participation in median number of d and interquartile interval overall and during the pandemic phases, in women eventually participating in the screening programme.

Using a generalised linear model with log link for the Poisson family with robust SE, we estimated prevalence ratios (PRs) and 95% CI of participation in cervical cancer screening within 90 d, 180 d, and 365 d, respectively, among women invited to participate in screening during the different phases of the pandemic overall and stratifying by the explanatory variables. Firstly, we calculated unadjusted analyses. Thereafter, the analyses were adjusted for month of invitation to allow for seasonality and year of invitation to take into account the underlying decreasing trend in participation in cervical cancer screening (*Regionernes Kliniske Kvalitetsudviklingsprogram, 2022*). Finally, the analyses were adjusted for age to take into account the effect of age on the other explanatory variables. Furthermore, to take into account the effect of time on each of the socio-economic variables, we performed an interaction test between time and each of the socio-economic variables using a Wald test. These tests for interaction were statistically significant; however, to allow for interpretation of the estimates within each strata of the socio-economic variables, a stratifying approach was used.

All analyses were conducted using STATA version 17.0.

## Ethical considerations

The study is registered at the Central Denmark Region's register of research projects (journal number 1-16-02-381-20). Patient consent is not required by Danish law for register-based studies.

# Results

## Descriptive characteristics of the study population

Altogether, 2,220,000 invited women (in 1,466,353 individuals) were included in the study. The median age at invitation was 40 y (interquartile range (IQR) = 30–49 y), the majority of women (82.2%) were of Danish descent, 45.9% were married, and 60.4% of women had a low educational level. The distribution of the descriptive characteristics was broadly similar throughout the study period (*Table 1*).

## Participation during the COVID-19 pandemic

*Figure 1* shows the participation in cervical cancer screening within 90, 180, and 365 d throughout the study period. Before the pandemic, approximately 36% of women participated in cervical cancer screening within 90 d, 54% of women participated within 180 d, and 65% of women participated within 365 d (*Supplementary files 1–3*).

In March and April 2020, the participation in cervical cancer screening within 90 d dropped markedly to approximately 20% after which the participation resumed to normal levels (*Figure 1*). This was also reflected in a PR of 0.58 (95% CI: 0.56–0.59) during pre-lockdown and a PR of 0.76 (95% CI: 0.75–0.77) during first lockdown, resuming to PRs of 0.96–0.99 throughout the rest of the study period (*Supplementary file 4*).

A reduction in the participation in cervical cancer screening within 180 d was also observed among women invited at the start of the pandemic (*Figure 2*) reflected in a PR of 0.89 (95% CI: 0.88–0.90) during pre-lockdown and a PR of 0.92 (95% CI: 0.91–0.93) during first lockdown. From first re-opening and onwards, the level of participation within 180 d returned to pre-pandemic levels (*Supplementary file 5*).

The participation in cervical cancer screening within 365 d among women invited at the early phase of the pandemic was only slightly reduced (*Figure 1*), reflected in overall PRs of 0.97 (95% CI: 0.96–0.98) during both pre-lockdown and first lockdown where after the participation increased to the same level as before the pandemic (*Table 2*).

**Table 1.** Baseline characteristics of women invited to participate in cervical cancer screening in Denmark from 2015 to 2021.

| | Pre-pandemic (01 January 2015–31 January 2020) | Pre-lockdown (01 February 2020–10 March 2020) | First lockdown (11 March 2020–15 April 2020) | First re-opening (16 April 2020–15 December 2020) | Second lockdown (16 December 2020–27 February 2021) | Second re-opening (28 February 2021–30 September 2021) | Total |
|---|---|---|---|---|---|---|---|
| | N (%) | N (%) | N (%) | N (%) | N (%) | N (%) | N (%) |
| Total | 1,641,199 (73.9) | 41,876 (1.9) | 31,255 (1.4) | 223,386 (10.1) | 69,729 (3.1) | 212,555 (9.6) | 2,220,000 (100.0) |
| **Age at invitation** | | | | | | | |
| 23–29 y | 384,272 (23.4) | 10,223 (24.4) | 7731 (24.7) | 58,569 (26.2) | 17,624 (25.3) | 56,816 (26.7) | 535,235 (24.1) |
| 30–39 y | 412,249 (25.1) | 10,614 (25.3) | 7712 (24.7) | 56,783 (25.4) | 17,571 (25.2) | 54,367 (25.6) | 559,296 (25.2) |
| 40–49 y | 495,153 (30.2) | 12,690 (30.3) | 9233 (29.5) | 63,872 (28.6) | 19,051 (27.3) | 59,366 (27.9) | 659,365 (29.7) |
| 50–59 y | 246,814 (15.0) | 6665 (15.9) | 5269 (16.9) | 34,841 (15.6) | 11,810 (16.9) | 31,387 (14.8) | 336,786 (15.2) |
| 60–64 y | 102,711 (6.3) | 1684 (4.0) | 1310 (4.2) | 9321 (4.2) | 3673 (5.3) | 10,619 (5.0) | 129,318 (5.8) |
| Median (IQI) | 41 (31; 49) | 40 (30; 48) | 40 (30; 49) | 39 (30; 48) | 40 (30; 49) | 39 (30; 48) | 40 (30; 49) |
| **Ethnicity** | | | | | | | |
| Danish descent | 1,358,106 (82.8) | 33,975 (81.2) | 25,778 (82.5) | 177,501 (79.5) | 45,340 (81.3) | 132,376 (81.0) | 1,773,076 (82.2) |
| Descendant of immigrant | 31,339 (1.9) | 918 (2.2) | 781 (2.5) | 5723 (2.6) | 1260 (2.3) | 3881 (2.4) | 43,902 (2.0) |
| Western immigrant | 87,100 (5.3) | 2581 (6.2) | 1655 (5.3) | 15,869 (7.1) | 3247 (5.8) | 9710 (5.9) | 120,162 (5.6) |
| Non-western immigrant | 163,638 (10.0) | 4375 (10.5) | 3027 (9.7) | 24,176 (10.8) | 5928 (10.6) | 17,535 (10.7) | 218,679 (10.1) |
| **Cohabitation status** | | | | | | | |
| Living alone | 529,023 (32.3) | 13,708 (32.8) | 10,300 (33.0) | 76,687 (34.4) | 3867 (35.1) | N/A | 633,585 (32.5) |
| Cohabiting | 351,991 (21.5) | 9225 (22.1) | 6829 (21.9) | 50,379 (22.6) | 2369 (21.5) | N/A | 420,793 (21.6) |
| Married | 759,009 (46.3) | 18,890 (45.2) | 14,088 (45.1) | 96,132 (43.1) | 4794 (43.5) | N/A | 892,913 (45.9) |
| **Educational level (ISCED)** | | | | | | | |
| ISCED15 level 1–2 | 960,324 (60.6) | 24,481 (59.3) | 18,667 (60.6) | 129,791 (59.3) | 40,565 (60.1) | 122,230 (60.4) | 1,296,058 (60.4) |
| ISCED15 level 3–5 | 393,390 (24.8) | 10,185 (24.7) | 7539 (24.5) | 52,354 (23.9) | 16,772 (24.8) | 48,870 (24.1) | 529,110 (24.7) |
| ISCED15 level 6–8 | 231,157 (14.6) | 6589 (16.0) | 4600 (14.9) | 36,716 (16.8) | 10,173 (15.1) | 31,395 (15.5) | 320,630 (14.9) |
| **Disposable income** | | | | | | | |
| Lowest quintile | 322,307 (19.9) | 7419 (18.2) | 5869 (19.0) | 43,920 (20.4) | 12,715 (18.6) | 39,486 (19.3) | 431,716 (19.8) |
| Second quintile | 334,113 (20.6) | 7611 (18.7) | 5878 (19.0) | 40,851 (19.0) | 12,137 (17.8) | 36,480 (17.9) | 437,070 (20.0) |
| Third quintile | 338,563 (20.9) | 8066 (19.8) | 5850 (18.9) | 40,863 (19.0) | 11,738 (17.2) | 34,157 (16.7) | 439,237 (20.1) |
| Fourth quintile | 326,174 (20.1) | 8604 (21.1) | 6449 (20.9) | 43,279 (20.1) | 14,025 (20.5) | 40,367 (19.8) | 438,898 (20.1) |
| Highest quintile | 301,566 (18.6) | 9067 (22.2) | 6878 (22.2) | 46,484 (21.6) | 17,742 (26.0) | 53,702 (26.3) | 435,439 (20.0) |
| **Healthcare usage** | | | | | | | |
| Rare | 319,960 (19.5) | 8519 (20.3) | 5985 (19.1) | 47,619 (21.3) | 13,413 (19.2) | 43,973 (20.7) | 439,469 (19.8) |
| Low | 366,807 (22.3) | 9020 (21.5) | 6774 (21.7) | 48,280 (21.6) | 15,181 (21.8) | 45,102 (21.2) | 491,164 (22.1) |
| Average | 347,589 (21.2) | 8758 (20.9) | 6637 (21.2) | 46,608 (20.9) | 14,364 (20.6) | 44,024 (20.7) | 467,980 (21.1) |
| High | 299,358 (18.2) | 7583 (18.1) | 5727 (18.3) | 40,091 (17.9) | 12,967 (18.6) | 38,760 (18.2) | 404,486 (18.2) |
| Frequent | 307,485 (18.7) | 7996 (19.1) | 6132 (19.6) | 40,788 (18.3) | 13,804 (19.8) | 40,696 (19.1) | 416,901 (18.8) |

*Table 1 continued on next page*

*Table 1 continued*

| | Pre-pandemic (01 January 2015–31 January 2020) | Pre-lockdown (01 February 2020–10 March 2020) | First lockdown (11 March 2020–15 April 2020) | First re-opening (16 April 2020–15 December 2020) | Second lockdown (16 December 2020–27 February 2021) | Second re-opening (28 February 2021–30 September 2021) | Total |
|---|---|---|---|---|---|---|---|
| | N (%) | N (%) | N (%) | N (%) | N (%) | N (%) | N (%) |
| Time from invitation to participation, median (IQI) | 94 (42; 200) | 120 (72; 207) | 122 (53; 201) | 86 (36; 161) | 69 (29; 133) | 51 (28; 101) | 89 (39; 184) |

IQI = interquartile interval; ISCED = International Standard Classification of Education; Information on cohabitation status was only available from Statistics Denmark until the end of February 2021, whereas all other socio-economic variables were available until end of the study period.

## Participation during the COVID-19 pandemic by socio-economic variables

Before the pandemic, the participation in cervical cancer screening within 365 d was lowest among the youngest age group (57%), among immigrants (44–50% in immigrants and 38% in descendants of immigrants), among women living alone (56%), among women with the lowest income level (52%), and among women who rarely use the healthcare system (52%; *Supplementary files 1-3*).

During pre-lockdown and first lockdown, the participation in screening within 365 d was reduced among women aged 40–49 y old, 60–64 y old, among descendants of immigrants, among women with a low educational level, and a low income (*Table 2*).

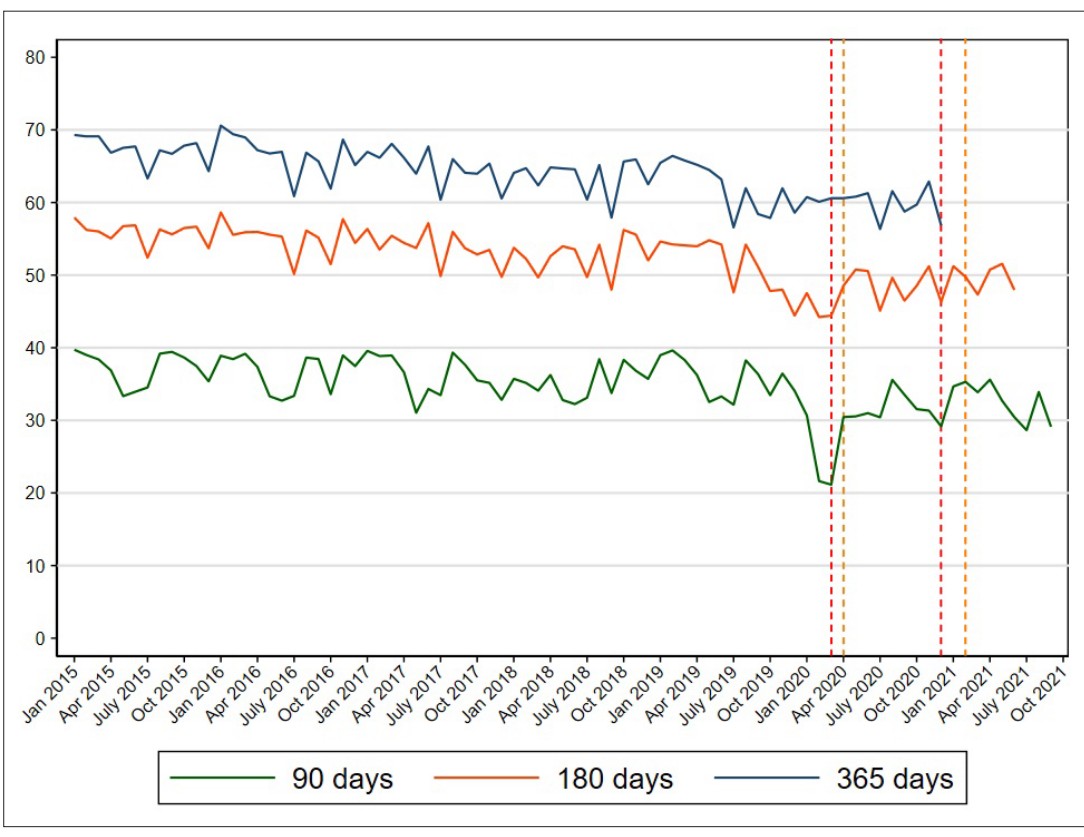

**Figure 2.** Participation in cervical cancer screening in Denmark within 90, 180, and 365 d since invitation from 2015 to 2021.

**Table 2.** Prevalence ratios and 95% CIs of participation in cervical cancer screening in Denmark within 365 d since invitation from 2015–2021*.

| | N | Pre-pandemic (01 January 2015–31 January 2020) N=164,1199 | | Pre-lockdown (01 February 2020–10 March 2020) N=41,876 | | First lockdown (11 March 2020–15 April 2020) N=31,255 | | First re-opening (16 April 2020–15 December 2020) N=223,386 | | Second lockdown (16 December 2020–31 December 2020) N=69,729 | |
|---|---|---|---|---|---|---|---|---|---|---|---|
| | | PR | (95% CI) | PR | (95% CI) | PR | (95% CI) | PR | (95% CI) | PR | (95% CI) |
| Overall | 2,220,000 | 1.00 | - | 0.97 | (0.96; 0.98) | 0.97 | (0.96; 0.98) | 1.01 | (1.00; 1.01) | 0.99 | (0.97; 1.00) |
| **Age at invitation** | | | | | | | | | | | |
| 23–29 y | 535,235 | 1.00 | - | 0.96 | (0.94; 0.98) | 0.98 | (0.95; 1.00) | 0.99 | (0.98; 1.00) | 1.00 | (0.96; 1.04) |
| 30–39 y | 559,296 | 1.00 | - | 0.97 | (0.95; 0.99) | 0.99 | (0.97; 1.01) | 1.01 | (1.00; 1.02) | 1.00 | (0.97; 1.04) |
| 40–49 y | 659,365 | 1.00 | - | 0.95 | (0.93; 0.96) | 0.94 | (0.92; 0.95) | 1.00 | (0.99; 1.01) | 0.93 | (0.90; 0.96) |
| 50–59 y | 336,786 | 1.00 | - | 1.01 | (0.99; 1.03) | 1.00 | (0.98; 1.02) | 1.02 | (1.01; 1.03) | 1.05 | (1.01; 1.08) |
| 60–64 y | 129,318 | 1.00 | - | 0.95 | (0.92; 0.99) | 0.93 | (0.90; 0.97) | 0.98 | (0.96; 1.00) | 0.93 | (0.87; 0.99) |
| **Ethnicity** | | | | | | | | | | | |
| Danish descent | 1,773,076 | 1.00 | - | 0.96 | (0.95; 0.97) | 0.96 | (0.95; 0.97) | 1.00 | (1.00; 1.01) | 0.97 | (0.96; 0.99) |
| Descendant of immigrant | 43,902 | 1.00 | - | 0.95 | (0.86; 1.05) | 0.88 | (0.79; 0.98) | 0.97 | (0.92; 1.02) | 0.93 | (0.78; 1.12) |
| Western Immigrant | 120,162 | 1.00 | - | 1.03 | (0.98; 1.09) | 1.04 | (0.98; 1.11) | 1.09 | (1.06; 1.12) | 1.06 | (0.95; 1.17) |
| Non-western immigrant | 218,679 | 1.00 | - | 0.99 | (0.95; 1.02) | 0.98 | (0.94; 1.02) | 1.04 | (1.02; 1.06) | 1.07 | (1.00; 1.14) |
| **Cohabitation status** | | | | | | | | | | | |
| Living alone | 633,585 | 1.00 | - | 0.97 | (0.95; 0.98) | 0.96 | (0.94; 0.98) | 1.02 | (1.01; 1.02) | 0.98 | (0.95; 1.02) |
| Cohabiting | 420,793 | 1.00 | - | 0.96 | (0.94; 0.98) | 0.97 | (0.95; 0.99) | 1.00 | (0.99; 1.01) | 1.00 | (0.96; 1.04) |
| Married | 892,913 | 1.00 | - | 0.97 | (0.96; 0.98) | 0.97 | (0.96; 0.99) | 1.00 | (1.00; 1.01) | 0.98 | (0.96; 1.01) |
| **Educational level (ISCED)** | | | | | | | | | | | |
| ISCED15 level 1–2 | 1,297,050 | 1.00 | - | 0.96 | (0.95; 0.97) | 0.96 | (0.95; 0.98) | 1.01 | (1.00; 1.01) | 0.99 | (0.97; 1.01) |
| ISCED15 level 3–5 | 529,165 | 1.00 | - | 0.96 | (0.95; 0.98) | 0.97 | (0.95; 0.99) | 1.00 | (0.99; 1.01) | 0.97 | (0.94; 1.00) |
| ISCED15 level 6–8 | 319,925 | 1.00 | - | 1.00 | (0.98; 1.02) | 0.99 | (0.97; 1.02) | 1.04 | (1.03; 1.05) | 1.06 | (1.01; 1.10) |
| **Disposable income** | | | | | | | | | | | |
| Lowest quintile | 419,122 | 1.00 | - | 0.95 | (0.93; 0.98) | 0.96 | (0.93; 0.98) | 1.02 | (1.01; 1.04) | 1.02 | (0.97; 1.07) |
| Second quintile | 422,225 | 1.00 | - | 0.95 | (0.92; 0.97) | 0.94 | (0.91; 0.96) | 1.00 | (0.99; 1.01) | 1.00 | (0.95; 1.04) |
| Third quintile | 424,081 | 1.00 | - | 0.96 | (0.94; 0.98) | 0.95 | (0.93; 0.97) | 0.99 | (0.98; 1.00) | 0.94 | (0.91; 0.98) |
| Fourth quintile | 425,069 | 1.00 | - | 0.96 | (0.95; 0.98) | 0.96 | (0.95; 0.98) | 1.00 | (0.99; 1.01) | 0.96 | (0.93; 0.99) |
| Highest quintile | 424,457 | 1.00 | - | 0.98 | (0.96; 0.99) | 0.96 | (0.95; 0.98) | 1.00 | (0.99; 1.00) | 0.97 | (0.95; 1.00) |
| **Healthcare usage** | | | | | | | | | | | |
| Rare | 439,469 | 1.00 | - | 0.97 | (0.95; 0.99) | 0.97 | (0.95; 1.00) | 1.02 | (1.01; 1.03) | 0.99 | (0.94; 1.04) |
| Low | 491,164 | 1.00 | - | 0.97 | (0.95; 0.99) | 0.95 | (0.93; 0.97) | 1.00 | (0.99; 1.01) | 0.95 | (0.91; 0.99) |
| Average | 467,980 | 1.00 | - | 0.95 | (0.94; 0.97) | 0.95 | (0.93; 0.97) | 1.00 | (0.99; 1.01) | 1.00 | (0.97; 1.04) |
| High | 404,486 | 1.00 | - | 0.97 | (0.95; 0.98) | 0.96 | (0.94; 0.98) | 1.01 | (1.00; 1.02) | 0.99 | (0.96; 1.03) |
| Frequent | 416,901 | 1.00 | - | 0.97 | (0.95; 0.99) | 0.98 | (0.97; 1.00) | 1.00 | (0.99; 1.01) | 0.96 | (0.93; 1.00) |

*Adjusted for month, year, and age at invitation; PR = prevalence ratio; ISCED = International Standard Classification of Education.

## Time to participation

The median time from invitation to participation was 94 d (IQR = 42–200) before the pandemic; however, this increased to 120 d (IQR = 72–207) among women invited during pre-lockdown and to 122 d (IQR = 53–201) during first lockdown. Thereafter, the time to participation resumed to 86 d during the first re-opening (*Table 1*).

## Discussion

### Main findings

In this population-based study, comprising 2,220,000 women invited for cervical cancer screening from 2015 to 2021 (in 1,466,353 individuals), we found a large decline in participation within 90 d since invitation during the early phase of the pandemic, a smaller decline in participation within 180 d, and only a slight reduction in participation within 365 d. The reduction in participation within 365 d was most pronounced among descendants of immigrants, among women with a low educational level, and a low income.

### Comparison with previous studies and explanation of findings

In most countries, population-based screening for cervical cancer was halted at the start of the pandemic. This led to pronounced reductions in the number of women screened for cervical cancer during the early phase of the pandemic (*Castanon et al., 2022*; *Cancer Registry of Norway, 2020*; *Meggetto et al., 2021*). To our knowledge, no studies have described the long-term participation in cervical cancer screening during the pandemic. We found a marked reduction (42% in pre-lockdown and 24% in first lockdown) in the short term (within 90 d) cervical cancer screening participation at the start of the pandemic compared with the previous years. This reduction in participation could be explained either by a change in health behaviour or could perhaps reflect inconsistent messages from the health authorities at the start of the pandemic. The screening programme was open, and invitations and reminders were sent out; however, at the same time, the health authorities asked the population to stay at home at the national televised press conferences, and simultaneously, the College of General Practitioners recommended general practitioners to postpone routine cervical screening samples during a 4 wk period in March/April 2020 (*Dansk Selskab for Almen Medicin, 2020*). The inconsistent health messages could thus have led women to not participate in screening. Congruently, a Danish qualitative study found that inconsistent health communication from the authorities led women to postpone or cancel their screening appointments (*Kirkegaard et al., 2021*). With the longer follow-up time, we observed a less reduced participation (11% in pre-lockdown and 8% in first lockdown within 180 d and only 3% in both pre-lockdown and first lockdown within 365 d), which was reflected by the longer time to participation (>120 d versus approximately 89 d) at the early phase of the pandemic. The disruption to the cervical cancer screening programme in Denmark thus appear only to have a temporary effect with most women resuming cervical cancer screening with a longer follow-up period. This is in accordance with findings in a qualitative study showing that women were concerned about visiting healthcare settings during the pandemic but were willing to participate when screening programmes resumed (*Wilson et al., 2021*). The marked reduction in participation in screening during pre-lockdown and first lockdown could thus also reflect fear of infection among women at the early phase of the pandemic. During second lockdown, no overall change in participation was seen, indicating that the population had been accustomed to navigating the healthcare system during the pandemic. In the Danish cervical cancer screening programme, reminders are mailed to non-participants after 3 and 6 mo, and this could have prompted women postponing or cancelling their screening appointments at the start of the pandemic to participate at a later time point. Furthermore, the general health communication from the authorities changed throughout the pandemic initially, asking the population to stay at home and then later on reminding the population to seek healthcare when needed.

The severity of the pandemic and the pandemic response varied across the world with Denmark managing to keep the number of hospitalisations due to COVID-19 at relatively low level (*Statens Serum Institut, 2021a*). The pandemic response in Denmark included periodic lockdowns, extensive COVID-19 testing free-of-charge to the whole population (*Pottegård et al., 2020*), and a high COVID-19 vaccination coverage. The cervical cancer screening participation may therefore be different in other countries with a different pandemic response and a more severe impact of the pandemic.

Women of lower socio-economic position (*Harder et al., 2018*) and immigrant women *Hertzum-Larsen et al., 2019*; *Badre-Esfahani et al., 2020* have earlier been shown to have a lower participation in cervical cancer screening. This was evident from our study also in that immigrants, women living alone, and women with a low-income level had the lowest participation in cervical cancer screening throughout the study period. A concern is that the pandemic may have affected socially disadvantaged individuals disproportionally. We found an overall 3% reduction in participation within 365 d;

however, among descendants of immigrants and among women with a low income, a 5% reduction was seen, and among women with a low educational level, a 4% reduction was found. It is therefore important to ensure that all women – regardless of socio-economic position – resume participation in cervical cancer screening at the aftermath of the pandemic. To our knowledge, our study is the first to describe cervical cancer screening participation during the pandemic according to socio-economic groups.

A few previous studies have examined the participation in cervical cancer screening during the pandemic according to age groups. One study found that women aged 30–39 y old (*Ivanuš et al., 2021*) had the lowest participation in screening during the first 6 mo of the pandemic, whilst another study showed that the oldest age groups (50–59 and 60–69 y old; *Walker et al., 2021*) had the lowest cervical cancer screening participation during the first year of the pandemic. Additionally, a study by Castañon et al. estimated that women aged 40–49 y old would have the greatest burden of excess cervical cancer diagnoses due to a delay in screening because of the pandemic (*Castanon et al., 2022*). We found that women aged 40–49 y old and 60–64 y old had a lower than usual participation in cervical cancer screening at the start of the pandemic. The pandemic thus appears to affect different age groups differently; however, this finding could also be due to chance. Women aged 60–64 y old may have been hesitant to come into contact with the healthcare system because of fear of infection, possibly explaining the lower participation in this age group. Older individuals and individuals with underlying health conditions have a higher risk of a worse outcome if exposed to COVID-19 and the level of comorbidity increases with age, which could explain the lower participation in the oldest age group because of fear of infection. Surprisingly, this effect lasted even when examining participation within 365 d since invitation. A concern is therefore that some women did not resume screening even with the longest follow-up time. The lower participation among women aged 40–49 y old cannot be easily explained and could thus be due to chance.

## Strengths and limitations

A major strength of the study is the high quality of data covering the entire population of women invited to participate in the cervical screening programme in Denmark. Danish national registers are known to be reliable and to have high completeness (*Thygesen et al., 2011*), which also confers to the Cervical Cancer Screening Database (*Rygaard, 2016*). While the quality of the Danish registers is high, some limitations relate to the data, for example, the study did not include data on comorbidities, which may affect participation in screening during the pandemic as individuals with underlying disease where advised to self-isolate at the height of the pandemic. However, as age is strongly associated with the level of comorbidity, the inclusion of age in the statistical model reduces the theoretical impact of comorbidity on the results.

## Implications of the findings

Our findings show that the overall participation in cervical cancer screening was almost at the same level as the previous years when allowing for the longest follow-up time; however, some groups had slightly lower participation (descendants of immigrants, women with a low educational level, and women with a low income), and it is therefore important to ensure that all women re-enter the cervical cancer screening programme at the aftermath of the pandemic. Our results also show that some age groups (women aged 40–49 y old and 60–64 y old) had a lower participation in screening than usual, possibly indicating that the restrictions within a society affects different age groups disproportionally, although this finding may be due to chance. It is thus important to take this information into account when planning a pandemic response and ensure that all women have access to screening.

Contrasting health messages may have been conveyed by the cervical cancer screening programme being open, the general practitioners recommending a postponement of cervical cancer screening tests, and at the same time, the health authorities recommending people to stay at home. Inconsistent health communication from the authorities may therefore have led some women to refrain from participating in screening. The health communication therefore needs to be precise and consistent.

## Conclusion

The cervical cancer screening programme continued throughout the COVID-19 pandemic in Denmark. The participation was reduced at the early phase of the pandemic; however, most women resumed

screening with the longest follow-up time, although women of lower socio-economic position had slightly lower participation than usual.

## Acknowledgements

We would like to thank Flemming Bro, MD, PhD, GP, Senior researcher from the Research Unit for General Practice, Aarhus for his valuable comments to the manuscript.

## Additional information

### Funding

| Funder | Grant reference number | Author |
|---|---|---|
| The Danish Cancer Society | R321-A17417 | Tina Bech Olesen |
| The Danish regions | | Tina Bech Olesen |

The funders had no role in study design, data collection and interpretation, or the decision to submit the work for publication.

### Author contributions

Tina Bech Olesen, Conceptualization, Data curation, Funding acquisition, Methodology, Writing – original draft, Writing – review and editing; Henry Jensen, Conceptualization, Data curation, Formal analysis, Methodology, Writing – review and editing; Henrik Møller, Jens Winther Jensen, Marianne Waldstrøm, Conceptualization, Funding acquisition, Writing – review and editing; Berit Andersen, Conceptualization, Supervision, Funding acquisition, Writing – review and editing

### Author ORCIDs

Tina Bech Olesen http://orcid.org/0000-0002-6295-7399
Henry Jensen http://orcid.org/0000-0003-4040-7334

### Ethics

Human subjects: Ethical considerations The study is registered at the Central Denmark Region's register of research projects (journal number 1-16-02-381-20). Patient consent is not required by Danish law for register-based studies.

### Decision letter and Author response

Decision letter https://doi.org/10.7554/eLife.81522.sa1
Author response https://doi.org/10.7554/eLife.81522.sa2

## Additional files

### Supplementary files

• Supplementary file 1. Proportion of women participating in cervical cancer screening in Denmark within 90 d since invitation from 2015 to 2021.

• Supplementary file 2. Proportion of women participating in cervical cancer screening in Denmark within 180 d since invitation from 2015 to 2021.

• Supplementary file 3. Proportion of women participating in cervical cancer screening in Denmark within 365 d since invitation from 2015 to 2021.

• Supplementary file 4. Prevalence ratios and 95% CIs of participation in cervical cancer screening in Denmark within 90 d since invitation from 2015 to 2021.

• Supplementary file 5. Prevalence ratios and 95% CIs of participation in cervical cancer screening in Denmark within 180 d since invitation from 2015 to 2021.

• MDAR checklist

## Data availability

Data availability statement In order to comply with the Danish regulations on data privacy, the datasets generated and analysed during this project are not publicly available as the data are stored and maintained electronically at Statistics Denmark, where it only can be accessed by pre-approved researchers using a secure VPN remote access. Furthermore, no data at a personal level nor data not exclusively necessary for publication are allowed to be extracted from the secure data environment at Statistics Denmark. Access to the data can; however, be granted by the authors of the present study upon a reasonable scientific proposal within the boundaries of the present project and for scientific purposes only.

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
