## [Editor Report]

This article shows how the COVID-19 pandemic affected cervical cancer screening participation in the organized screening program of Denmark. Through registry data covering the entire population, the study shows that while short-term (90 days) participation after invitation dropped, long-term (365 days) participation remained stable. These results will be of interest to public health specialists and researchers working on pandemic recovery efforts related to cancer screening worldwide.

---

## [Decision Letter]

**Decision letter after peer review:**

Thank you for submitting your article "Participation in the nation-wide cervical cancer screening programme in Denmark during the COVID-19 pandemic: An observational study" for consideration by *eLife*. Your article has been reviewed by 2 peer reviewers, one of whom is a member of our Board of Reviewing Editors, and the evaluation has been overseen by Diane Harper as the Senior Editor. The following individual involved in the review of your submission has agreed to reveal their identity: Maarit Leinonen (Reviewer #2).

Essential revisions:

1) Please consider including a formal test of the time*socioeconomic variable interaction to address both reviewers' concerns regarding the role of chance when seeing differences of effect across groups.

2) Address requests for clarifications in methods by reviewers below.

3) Consider discussing the role that fear may have played in determining cervical cancer screening participation during the pandemic.

*Reviewer #1 (Recommendations for the authors):*

Overall great study, I have very few comments:

• Please consider having the 90-day results as the main outcome presented in the main text for the model regression; I think these are most informative as the potentially highest impact that COVID-19 can have on screening participation.

• Please consider including a formal test of the time*socioeconomic variable interaction in the Poisson regression to assess whether there are differences between groups of women in their screening participation over time. This would allow for example assessing whether there is good support for women of different ages being differently impacted by the pandemic in their screening participation.

• It would be worth mentioning if there was any effort made by the screening program to incite women to screen which would explain why 90-day participation dropped but 365-day participation did not, ex. were any reminders sent to women who do not participate by a certain time or were there any public health messaging campaigns?

• For the median and IQI time from invitation to participation, it is not clear whether this statistic is calculated among all women invited, or only among the invited women who eventually participated in screening (excludes women who do not respond to invitation). Please clarify.

*Reviewer #2 (Recommendations for the authors):*

I recommend publishing this paper, but I have a few concerns which I summarize here.

My major concern is the study methods which are occasionally a bit hard to follow. The authors write that the study population comprised all screening-aged women from 1 January 2015 to 30 September 2021. Thus, was data on cervical cytology samples from the Danish Pathology Register extracted up to 30 September 2020? Also, the exposure of interest was the COVID-pandemic is a bit unclear. I assume that exposure is invitation to screening during the pre-pandemic period and COVID-pandemic. If that is the case, authors should make it clear that time periods refer to the time on invitation and not the time of outcome i.e. cervical cancer screening test. Also, the authors write that pre-lockdown and 1st lockdown was the start of the pandemic. There cannot be two starting points unless there are sensitivity analyses in which the onset varies. Thus, define clearly what is the starting point 1st February or 11 March 2020. In supplementary tables 1 to 3 time period for 2nd lockdown varies which is confusing. Please clarify the periods for exposure, outcome, and covariates.

Authors write that women who unregistered from the screening programme within 1 year since invitation (n=56,920) were excluded. Is there any information on who are these women and what are the reasons for unregistration? If those who are at higher risk of cancer and with lower participation rates unregister themselves, the compliance to screening could be overestimated.

Authors find that some age groups i.e. women aged 40-49 and those aged 60-64 years had a lower participation rate and conclude that it could indicate that the restrictions within a society affect different age groups disproportionally. The authors do not try to explain the finding and it should be scrutinized to rule out a chance. Comorbidity is strongly associated with age so if this is attributed to self-isolation, there should be a gradient. Why 50-59 years old would be different from 60-64 years? Have e.g. possible interactions between demographic and socioeconomic variables been taken into account in the analyses? The number of average health care visits 7-11 visits per year seems extremely high average for the mainly working-age population which in general is quite healthy. How these categories were decided? A priori or after exploring the data? If former, is there a reference that provides information on the average use of health care services?

Authors have cited the work by Wilson et al. (Ref # 22) in which only 4.1% of respondents were worried about catching coronavirus if they went for screening. Authors could add some discussion and references concerning fear and overall healthcare utilization during the early phase of the pandemic.

lines 73-76 authors write that prioritisations and re-organisations were done within the healthcare system to ensure the capacity to take care of patients in need of hospitalization due to COVID-19. While in Denmark laymen were trained for COVID-19 sample taking, in many other countries health care personnel were needed for sample taking and laboratory analytics. Thus, the possible lack of resources in screening programmes was not only due to hospitalized patients.

lines 204-205: Authors have adjusted for the year of invitation due to decreasing trend in screening uptake. Is there a reference for this? Any explanations for the trend? Could it be, for instance, increasing diversity in the screening population?

lines: 360-361: Authors write that women should be well-informed when they can safely participate in cervical cancer screening during the pandemic. When it is safe to participate? What is the definition of safe participation? Service providers can of course do risk mitigation interventions but who can guarantee that nobody will ever catch an infection? Consider rephrasing.

Table 1. Why the cohabitation status is not available for the latest period (2nd re-opening)?

Supplementary Figures. I assume that the authors want to demonstrate monthly variations and the dip in 1st lockdown. That is already provided in Figure 1. The same pattern is seen in all covariates and categories so perhaps not needed to repeat here. Curves are hard to interpret. There is an exhaustive list of supplementary material, that does not seem to provide any important value for the paper.

---

## [Author Response]

Essential revisions:1) Please consider including a formal test of the time*socioeconomic variable interaction to address both reviewers' concerns regarding the role of chance when seeing differences of effect across groups.

Thank you for raising the relevant point of interaction. Initially we opted to address the possible effect of interaction by using a stratifying approach. This choice was taken to ease the readability and interpretation for a wide range of readers. However, we do acknowledge that some readers will prefer also to see an explicit statistical test for interaction and we have therefore tested for all interactions between time*socioeconomic variables. These tests are all statistically significant.

Still, that to ease the readability and interpretation, we argue that we will keep only to report the estimates and confidence intervals for the stratified analyses, as this approach provides interpretable estimates. However, to accommodate the request from the reviewers, we have added in manuscript that we have undertaken these formal interaction test, and that these were statistically significant (page 6, lines 221-225).

2) Address requests for clarifications in methods by reviewers below.

We have clarified these requests below.

3) Consider discussing the role that fear may have played in determining cervical cancer screening participation during the pandemic.

Thank you for this comment. We have elaborated on the role of fear of infection in the Discussion section (page 9, lines 315-318 and page 10, line 354-361).

Reviewer #1 (Recommendations for the authors):Overall great study, I have very few comments:• Please consider having the 90-day results as the main outcome presented in the main text for the model regression; I think these are most informative as the potentially highest impact that COVID-19 can have on screening participation.

Thank you for this comment. We have opted to present the 365-day participation as the main outcome since we consider the long-term participation as the most important outcome of significance for the population. Cervical cancer screening is about prevention and a relatively short postponement of screening participation is assumed not to cause much harm. We therefore present the 90- and 180-day participation as supplementary tables.

• Please consider including a formal test of the time*socioeconomic variable interaction in the Poisson regression to assess whether there are differences between groups of women in their screening participation over time. This would allow for example assessing whether there is good support for women of different ages being differently impacted by the pandemic in their screening participation.

Thank you for this comment. Please see the response above.

• It would be worth mentioning if there was any effort made by the screening program to incite women to screen which would explain why 90-day participation dropped but 365-day participation did not, ex. were any reminders sent to women who do not participate by a certain time or were there any public health messaging campaigns?

Thank you for this comment. As stated in the Methods section (page 4 lines 117-119) and the Discussion section (page 9 lines 318-321) reminders to participate in cervical cancer screening were mailed as per routine to non-participants after 3 months and again after 6 months. No additional efforts were made at an individual level during the pandemic; however, the general health communication changed during the pandemic reminding the population to continue to seek healthcare. We have added this in the Discussion section (page 9, lines 321-324).

• For the median and IQI time from invitation to participation, it is not clear whether this statistic is calculated among all women invited, or only among the invited women who eventually participated in screening (excludes women who do not respond to invitation). Please clarify.

Thank you for this comment. We have clarified this sentence in the Methods section (page 6, lines 210-211), stating that time to participation was calculated among women eventually participating in the screening program.

Reviewer #2 (Recommendations for the authors):I recommend publishing this paper, but I have a few concerns which I summarize here.My major concern is the study methods which are occasionally a bit hard to follow. The authors write that the study population comprised all screening-aged women from 1 January 2015 to 30 September 2021. Thus, was data on cervical cytology samples from the Danish Pathology Register extracted up to 30 September 2020?

Thank you for this comment. To ensure at least 90 days complete follow-up on all tests, the data on cervical cytology samples covered up until 31 December 2021. We have added this in the Methods section (page 5, lines 165-170).

Also, the exposure of interest was the COVID-pandemic is a bit unclear. I assume that exposure is invitation to screening during the pre-pandemic period and COVID-pandemic. If that is the case, authors should make it clear that time periods refer to the time on invitation and not the time of outcome i.e. cervical cancer screening test.

Thank you for this comment. We have clarified this in the Methods section (page 5, lines 175-177).

Also, the authors write that pre-lockdown and 1st lockdown was the start of the pandemic. There cannot be two starting points unless there are sensitivity analyses in which the onset varies. Thus, define clearly what is the starting point 1st February or 11 March 2020.

Thank you for this comment. We have clarified this by using the terms pre-lockdown and 1^st^ lockdown throughout the manuscript instead.

In supplementary tables 1 to 3 time period for 2nd lockdown varies which is confusing. Please clarify the periods for exposure, outcome, and covariates.

Thank you for this comment. The time periods stated in the Supplementary Table 1-3 are correct. This is because of the difference in observation periods that is, participation in screening within 90 days (Supplementary Table 1), within 180 days (Supplementary Table 2) and within 365 days (Supplementary Table 3) since invitation. In Supplementary Table 3, the end of the inclusion period is 31 December 2020 to allow for 365 days of observation. We have clarified this in the Methods section (page 5, lines 165-170).

Authors write that women who unregistered from the screening programme within 1 year since invitation (n=56,920) were excluded. Is there any information on who are these women and what are the reasons for unregistration? If those who are at higher risk of cancer and with lower participation rates unregister themselves, the compliance to screening could be overestimated.

Thank you for this comment. Theoretically we do agree with this point. However, as these 56,920 women only constitutes 2.5% of all invited women, the theoretical bias to the participation rate (compliance to the programme) is minimal. However, as the paper focus on the relationship between the phases of the COVID-19 pandemic and participation rates in cervical cancer screening, for the risk of the number of unregistrations to be troublesome, it has to be associated with the phases of the pandemic. However, the women unregistered throughout both the reference period and the pandemic period, and thus the risk of bias is minimal. So in order to keep our already comprehensive paper as concise and readable as possible, we argue not to provide this to the manuscript as this cannot explain any of our findings.

Authors find that some age groups i.e. women aged 40-49 and those aged 60-64 years had a lower participation rate and conclude that it could indicate that the restrictions within a society affect different age groups disproportionally. The authors do not try to explain the finding and it should be scrutinized to rule out a chance. Comorbidity is strongly associated with age so if this is attributed to self-isolation, there should be a gradient. Why 50-59 years old would be different from 60-64 years? Have e.g. possible interactions between demographic and socioeconomic variables been taken into account in the analyses?

Thank you for this comment. We have elaborated on both age, comorbidity and chance findings in the Discussion section (page 10, line 353-363 and page 11, line 384). As stated in the Methods section, all analyses were adjusted for age to take into account the effect of age on the other explanatory variables (page 6, lines 219-221).

The number of average health care visits 7-11 visits per year seems extremely high average for the mainly working-age population which in general is quite healthy. How these categories were decided? A priori or after exploring the data? If former, is there a reference that provides information on the average use of health care services?

Thank you for this comment. As stated in the Methods section (page 6, lines 191-198), the number of healthcare visits comprise the total number of contacts to general practitioners, private practising medical specialists, physiotherapists, and chiropractors. It is understandable that 7-11 visits per year may seem quite high for a non-Danish reader. However, a woman aged 30-64 years in Denmark contacts their general practitioner (primary care physician) just over 8 times per year (reference in Danish: https://www.sdu.dk/da/sif/ugens_tal/04_2017). The reason these numbers are high is that healthcare contacts covers both face-to-face, telephone, and e-mail consultations in Denmark. The categories were made using the nearest number of contacts to 20th, 40th, 60th, and 80th percentiles in the data, thus the categories broadly reflects the quintiles. We have clarified this in the Methods section (pages 5-6, lines 191-198).

Authors have cited the work by Wilson et al. (Ref # 22) in which only 4.1% of respondents were worried about catching coronavirus if they went for screening. Authors could add some discussion and references concerning fear and overall healthcare utilization during the early phase of the pandemic.

Thank you for this comment. We have elaborated on the role of fear of infection in the Discussion section (page 9, lines 315-318 and page 10, line 354-361).

lines 73-76 authors write that prioritisations and re-organisations were done within the healthcare system to ensure the capacity to take care of patients in need of hospitalization due to COVID-19. While in Denmark laymen were trained for COVID-19 sample taking, in many other countries health care personnel were needed for sample taking and laboratory analytics. Thus, the possible lack of resources in screening programmes was not only due to hospitalized patients.

Thank you for this comment. We have clarified this in the Methods section (page 4, lines 132-134).

lines 204-205: Authors have adjusted for the year of invitation due to decreasing trend in screening uptake. Is there a reference for this? Any explanations for the trend? Could it be, for instance, increasing diversity in the screening population?

Thank you for this comment. We have added a reference to the annual report from the Cervical Cancer Screening Database in the Methods section (page 6, line 219), which on page 16 shows the decreasing participation in cervical cancer screening in Denmark. We have not elaborated on the reason for the decreasing participation in cervical cancer screening as this is beyond the scope of our study; however, this could be due to an increasing diversity in the population or due to the implementation of the HPV vaccination programme.

lines: 360-361: Authors write that women should be well-informed when they can safely participate in cervical cancer screening during the pandemic. When it is safe to participate? What is the definition of safe participation? Service providers can of course do risk mitigation interventions but who can guarantee that nobody will ever catch an infection? Consider rephrasing.

Thank you for this comment. We have re-phrased this sentence in the Discussion section (page 11, lines 392-394).

Table 1. Why the cohabitation status is not available for the latest period (2nd re-opening)?

Thank you for this comment. These data were unfortunately not available at Statistics Denmark at the time of the study, as this information is only updated yearly for research purposes. We have stated this in the Methods section (page 6, lines 200-202) and in a foot note to Table 1 and Supplementary Tables 1, 2, 4 and 5.

Supplementary Figures. I assume that the authors want to demonstrate monthly variations and the dip in 1st lockdown. That is already provided in Figure 1. The same pattern is seen in all covariates and categories so perhaps not needed to repeat here. Curves are hard to interpret. There is an exhaustive list of supplementary material, that does not seem to provide any important value for the paper.

Thank you for this comment. We have omitted Supplementary Figure 2-4.